# Prognostic Significance of Tumor Location in T2 Gallbladder Cancer: A Korea Tumor Registry System Biliary Pancreas (KOTUS-BP) Database Analysis

**DOI:** 10.3390/jcm9103268

**Published:** 2020-10-12

**Authors:** Seung Eun Lee, Yoo Shin Choi, Yong Hoon Kim, Jin Seok Heo, Chi-Young Jeong, Woo Jung Lee, Hyung Il Seo, Yoo-Seok Yoon, Jin-Young Jang

**Affiliations:** 1Department of Surgery, Chung-Ang University College of Medicine, Seoul 06973, Korea; selee508@cau.ac.kr (S.E.L.); choiys@cau.ac.kr (Y.S.C.); 2Department of Surgery, Division of Hepatobiliary and Pancreatic Surgery, Keimyung University Dongsan Medical Center, Daegu 42601, Korea; hbps@dsmc.or.kr; 3Department of Surgery, Samsung Medical Center, Sungkyunkwan University College of Medicine, Seoul 06351, Korea; jsheo@skku.edu; 4Department of Surgery, Gyeongsang National University College of Medicine, Jinju 52727, Korea; drjcy@hanmail.net; 5Department of Surgery, Yonsei University College of Medicine, Seoul 03722, Korea; wjlee@yumc.yonsei.ac.kr; 6Department of Surgery, Biomedical Research Institute, Pusan National University Hospital, Pusan 49241, Korea; seohi71@hanmal.net; 7Department of Surgery, Seoul National University Bundang Hospital, Seoul National University College of Medicine, Seoul 13620, Korea; yoonys@snubh.org; 8Department of Surgery and Cancer Research Institute, Seoul National University College of Medicine, Seoul 03080, Korea

**Keywords:** gallbladder, carcinoma, peritoneal, hepatic, prognosis

## Abstract

Background: T2 gallbladder cancer (GBC) is subdivided into T2a and T2b by the American Joint Committee on Cancer (AJCC) 8th edition. However; there is a lack of evidence for the prognostic significance of tumor location and validation with large-scale studies is needed. The aims of this study were to investigate the clinical features and clinical outcomes of T2 GBC according to tumor location and determine the prognostic significance of tumor location and an appropriate surgical strategy. Methods: Between 2000 and 2014 the Korea Tumor Registry System Biliary Pancreas (KOTUS-BP) database was used to identify and enroll a total 707 patients with pathologically diagnosed T2 GBC who underwent curative resection. Clinicopathological findings and long-term follow-up results were analyzed. Results: The incidence of lymph node metastasis in T2b was significantly higher than that of T2a tumors (37.9% vs. 29.5%, *p* = 0.032). The 5-year disease-specific survival of T2a was better than that of T2b tumors (74.8% vs. 65.4%, *p* = 0.019). There was no significant survival difference in T2a between extended cholecystectomy and simple cholecystectomy with lymph node dissection (81.8% vs. 73.7%, *p* = 0.361). However; there was a better survival trend for T2b tumor after extended cholecystectomy (71.7% vs. 59.3%, *p* = 0.057). Adjuvant chemotherapy was associated with improved survival for patients with lymph node metastasis in T2a (72.1% vs. 56.9; *p* = 0.022) and in T2b (68.2 vs. 48.5; *p* < 0.001). Multivariate analysis revealed that lymph node metastasis was the only significant poor prognostic factor (Hazard ratio 3.222; 95% confidential interval 1.960–4.489; *p* < 0.001). Conclusions: For T2 GBC; tumor location was not an independent prognostic factor. Lymph node metastasis was a significant poor prognostic factor and adjuvant chemotherapy should be considered for the patients with lymph node metastasis.

## 1. Introduction

Patients with T1 gallbladder cancer (GBC) generally have a good prognosis, whereas patients with advanced GBC such as T3 or T4 generally have a dismal prognosis. However, compared with T1, T3, and T4 GBC, the prognosis of T2 GBC is very heterogeneous and is difficult to predict. Recently, the heterogeneous prognosis of T2 GBC has been demonstrated to be related in part to its location; a T2 GBC on the peritoneal side has a better prognosis, while a tumor on the hepatic side has worse prognosis [1,2,3,4,5,6]. The newly published American Joint Committee on Cancer (AJCC), eighth edition has subdivided T2 GBC into two categories according to the location of the primary tumor: peritoneal side tumor (pT2a) and hepatic side tumor (pT2b) [7]. Since Shindoh and colleagues reported that GBC on the peritoneal side is associated with a higher five-year survival rate than that on the hepatic side [1], the superior prognosis of T2a over T2b GBC has been reproduced in several studies [2,3,4,5,6].

Although it is generally accepted that T2a GBC has better survival than T2b GBC, there is no consensus regarding the method used to define the tumor (radiological, pathological, combination) or the criteria defining tumor location. Further, there is lack of evidence regarding the clinicopathological differences that affect the differences in survival and the appropriate surgical strategy. Shindoh and colleagues defined the location of the tumor histopathologically [1], while the other studies defined the tumor location radiologically [3,4,5,6,7]. The anatomical difference between peritoneal and hepatic sides of the gallbladder may contribute to the differences that affect survival, though further study is needed to confirm this. The hepatic side of the gallbladder has no serosa, is attached directly to the liver by loose connective tissue and has dense lymphatic, arterial and venous communications which allow it to easily invade the liver. In contrast, the peritoneal side of the gallbladder is separated from the adjacent organs. Considering the surgical strategy, some authors recommend hepatic resection solely for hepatic side tumors and not for peritoneal side tumor [2,6], while others recommend hepatic resection for both hepatic side tumor and peritoneal side tumor [1,3].

Korea is one of the high-incidence countries for GBC; therefore, we investigated the clinical features and clinical outcomes of T2 GBC according to tumor location. We also determined the prognostic significance of tumor location and examined appropriate surgical strategy for T2 GBC using the Korea Tumor Registry System Biliary Pancreas (KOTUS-BP) database.

## 2. Experimental Section

### 2.1. Patients and Study Design

This study used the KOTUS-BP database to identify patients who were pathologically diagnosed with T2 GBC. This database was founded in 2015 by the Korean Association of Hepato-Biliary Pancreatic Surgery. To obtain actual survival data, we confined the study period to 1 January 2000–31 December 2014, which allowed the latest case to achieve five postoperative years. The Clinicopathological findings and long-term follow-up results were analyzed. This retrospective study conformed with the ethical guidelines of the Declaration of Helsinki and was approved by the investigational review board or ethics committee at each institute (C2016020-1763).

### 2.2. Tumor Stage

The tumor stage was determined, according to the AJCC Staging System, the Eighth Edition [7]. T2 GBC was defined as cancer confined to the perimuscular connective tissue with no extension beyond the serosa or into the liver. T2 GBC was subdivided into pT2a, defined as a T2 tumor located on the peritoneal side, and pT2b was defined as a T2 tumor located on the hepatic side. The extent of nodal disease was transformed into the categorical variables, NX (LN cannot be assessed), N0, N1 (one to three positive lymph nodes, LN) and N2 (four or more positive LN).

### 2.3. Tumor Location

To determine each tumor location, the operative findings, histopathologic reports and the images of computed tomography (CT) were collected separately besides registry data and were re-evaluated by two surgeons (LSE, CYS) who are specialized to biliary pancreas surgery more than 15 years of experience. If there was disagreement between two investigators, the location was determined through consultation with a radiologist who is specialized to hepato-biliary pancreas. The patients whose CT images were available and tumor location could be determined by these images were included in the present study. The tumor was classified as T2a if the entire tumor was on the peritoneal side of gallbladder. The tumor was classified as T2b if any portion of the tumor was located on the liver side of the gallbladder.

### 2.4. Operative Procedures

Operative procedures were defined as follows. A simple cholecystectomy (SCx) was defined as cholecystectomy alone, irrespective of LN dissection. Extended cholecystectomy (ECx) was defined as cholecystectomy, liver wedge resection, or segment 4b and 5 bisegmentectomy and regional LN dissection. Even for SCx, an achieved pathologically negative surgical margin was regarded as an R0 resection. 

### 2.5. Statistical Analysis

Continuous data are expressed as mean ±SD. Categorical variables were compared using Pearson’s chi-square test and continuous variables using the Mann–Whitney U test. All parameters with *p* < 0.05 by univariate analysis were included in the multivariate model. The overall survival time was calculated from the operation date to either the last follow-up date for surviving patients or the date of death due to GBC. Survival statuses and cause of death were confirmed with the assistance of the Korean Ministry of Public Administration and Security. To improve the completeness of the data, the KOTUS-BP database registry data manager visited the enrolled institutes and compiled the dissing data. Survival was calculated using the Kaplan–Meier method, and differences were analyzed using the log-rank test. A Cox regression model was used to identify the prognostic factors. Statistical analysis was performed using IBM SPSS software (version23.0 for Windows; IBM, Armonk, NY, USA).

## 3. Results

### 3.1. Demographics

A total 707 patients from whom images of computed tomography could be obtained to determine tumor location with pathologically diagnosed T2 GBC who underwent curative resection between 2000 and 2014 were identified from the KOTUS-BP database and included in this study. The male-to-female ratio was 1:1.3. The mean patient age at diagnosis was 66 years (range 25–92), and median follow-up period was 43 months (range 3.2–180). There were 310 patients with pT2a GBC (43.8%) and 397 patients with pT2b GBC (56.2%). Table 1 summarizes the patients’ clinicopathological characteristics according to tumor location.

### 3.2. Surgical Interventions and Postoperative Morbidity and Mortality

In T2a GBC, SCx was performed in 49 patients (15.8%), SCx with LN dissection was performed in 69 patients (22.3%) and ECx was performed in 192 patients (61.9%). In T2b GBC, SCx was performed in 59 patients (14.9%), SCx with LN dissection was performed in 65 patients (16.4%) and ECx was performed in 273 patients (68.8%). Fifty five patients (18.5%) underwent a second operation (liver resection with LN dissection) after SCx and there were no residual tumors at the gallbladder bed in all 55 resected specimens. Postoperative morbidities occurred in 88 patients (12.4%). There was no postoperative in-hospital mortality.

### 3.3. LN Metastasis

LN dissection was performed for 599 patients (84.7%). The overall mean number of retrieved LN was 8.5 (range 2–47), and 87 patients (16.6%) had fewer than three retrieved LNs. There was no significant difference in number of retrieved LN between T2a and T2b group (8.4 vs. 8.6, *p* = 0.664). LN metastasis occurred in 206 patients (206/599, 34.4%). Among them, 183 patients (88.8%) were N1 stage and 23 patients (11.2%) were N2 stage. The incidence of LN metastasis in T2b GBC was 37.9% (n = 128), which was significantly higher than that of the T2a tumor (n = 77, 29.5%, *p* = 0.032) (Table 1). 

### 3.4. Recurrence

During 43 months (range 3–189) of the median follow-up period, 70 patients (22.6%) with T2a tumor and 131 patients (33.0%) with T2b tumor experienced disease recurrence(*p* = 0.006). Systemic recurrence (n = 137, 71.3%) was more common than loco-regional recurrence (n = 56, 29.0%). The organ with most frequent recurrence was the liver (60/201, 29.9%). Among the patients with liver recurrence, eight had recurrence at the gallbladder bed; all had received ECx. Gallbladder bed recurrence occurred more frequently in T2b GBC (n = 7, 6.7%) than in T2a GBC (n = 1, 17.1%) (Table 2). Median time to recurrence was 10.4 months. Recurrence occurred more frequently in patients with T2b GBC (33.0% vs. 22.6%, *p* = 0.006), LN metastasis (54.8% vs. 24.8%, *p* = 0.001), moderate or poor differentiation (76.4% vs. 60.4%, *p* = 0.011), lymphovascular invasion (44.1% vs. 28.3%, *p* = 0.003) and perineural invasion (35.8% vs. 21.0%, *p* = 0.004).

### 3.5. Long-Term Survival

The 5-year disease-specific survival rate for the patients with T2a GBC was 74.8%, while that for patients with T2b GBC was 65.4% (*p* = 0.019) (Figure 1). After exclusion of 108 patients with Nx disease, 599 patients were included in a further sub-analysis. There was no significant difference in 5-year disease-specific survival rate between T2aN0 (Stage IIa) and T2bN0 (Stage IIb) (87.6% vs. 79.9%, *p* = 0.190) and between T2aN1 and T2bN1 (60.2% vs. 56.9%, *p* = 0.433). T2aN0 (Stage IIa) or T2bN0 (Stage IIb) showed significantly better survival than Stage IIIb (T2N1) (87.6%, 79.9% vs. 58.0%, *p* < 0.001). Comparing survival according to operation, there was no significant difference in 5-year disease-specific survival rate between ECx and SCx with LN dissection for T2a tumor (81.8% vs.73.7%, *p* = 0.361) (Figure 2a). However, there was a better survival trend without statistical significance for T2b tumor in ECx than in SCx+ LN dissection (71.7% vs. 59.3%, *p* = 0.057). (Figure 2b). Adjuvant chemotherapy was associated with improved overall survival only for patients with LN metastasis in T2a tumor (72.1% vs. 56.9, *p* = 0.022) and in T2b tumor (68.2 vs. 48.5, *p* < 0.001).

### 3.6. Prognostic Factors

Univariate analysis showed that T2b stage, LN metastasis, higher N stage, lymphovascular invasion, and perineural invasion were significant prognostic factors (Table 3). However, multivariate analysis revealed that LN metastasis was the only significant poor prognostic factor (hazard ratio (HR) 3.222, *p <* 0.001).

## 4. Discussion

The present study included a relatively large number of patients with GBC, in one of the most prevalent geographic areas for GBC, and revealed a significantly worse survival for pT2b GBC than in pT2a GBC (65.4% vs. 74.8%, *p* = 0.019). Furthermore, LN node metastasis (37.9% vs. 29.5%, *p* = 0.032), perineural invasion (36.9% vs. 19.7%, *p <* 0.001), and recurrence (33.0% vs. 22.6%, *p* = 0.006) was more frequent in T2b GBC than in T2a GBC. However, tumor location was not a significant prognostic factor in multivariate analysis. Subgroup analysis revealed that in terms of the presence of LN metastasis, there was no significant survival difference according to tumor location (T2aN0 vs. T2bN0, 87.6% vs. 79.9%, *p* = 0.190) (T2aN1 vs. T2bN1, 60.2% vs. 56.9%, *p* = 0.433). The fact that the incidence of LN metastasis in pT2b GBC (37.9%) was significantly higher than that of pT2a GBC (29.5%, *p* = 0.032) and LN metastasis was a significant prognostic factor in multivariate analysis, suggests that the prognostic significance of tumor location in T2 GBC might be not caused by the location itself, but by increased LN metastasis associated with the location. A hepatic side tumor showed a higher possibility of LN metastasis and thereby a worse prognosis than a peritoneal side tumor. All recent studies on T2 GBC have shown LN metastasis as a poor prognostic factor in T2 GBC, and a higher incidence of LN metastasis in pT2b GBC than pT2a GBC [1,2,3,4,5,6]. T2 GBC is generally known to have a high rate of LN metastasis, 33–62% despite its limited depth of invasion into the gallbladder wall [8,9,10,11,12]. It remains unclear why patients with pT2b tumor tend to have a higher incidence of LN metastasis than those with pT2a tumor. Perimuscular connective tissue contains more and larger lymphatic vessels than the shallower layers in the normal gallbladder [13,14], and the wall on the hepatic side of a normal gallbladder contains more lymphatic vessels than that on the peritoneal side [15]. Hepatic side of gallbladder is known to drain directly into an intra-hepatic venous or lymphatic route, while peritoneal side of gallbladder usually drain into the pericholecystic route [13,14,15]. These anatomical differences may partly explain why patients with pT2b GBC are more likely to have regional LN metastases than those with pT2a GBC. Further study to examine anatomical differences between T2a and T2b GBC should be performed to answer this question conclusively.

In the present study, ECx for T2b GBC showed a better survival trend without significant difference than SCx with LN dissection (71.7% vs. 59.3%, *p* = 0.057), but not for T2a GBC (81.8% vs. 73.7%, *p* = 0.361). For T2 GBC, published guidelines recommend ECx because partial hepatectomy is thought to be valuable in the aspect of achievement of tumor-free margin on the liver side and prophylactic resection to prevent liver metastasis [16,17,18,19]. Our results can be interpreted based on this hypothesis. The T2b GBC is located on the hepatic side and can spread to the liver without penetrating the serosa. Therefore, to achieve a tumor-free margin, gallbladder bed resection (ECx) is recommended. Although in the present study, no residual disease was detected in 25 patients who re-operated for T2b GBC, there might be selection bias because only 18.5% of patients underwent second operation and there was no consensual criteria for second operation in this retrospective multicenter study. In contrast, T2a GBC is located in the peritoneal side and is separated from the liver. Therefore, hepatectomy is not needed to achieve a negative resection margin in such situations. To determine whether different surgical strategies should be applied according to tumor location, a further well-designed prospective large-scale study should be performed.

In the present study, T2 GBC had a high incidence of LN metastasis, recurrence in patients with LN metastasis, and systemic recurrence after surgical resection. Furthermore, adjuvant chemotherapy significantly improved overall survival for the patients with LN metastasis in T2a (72.1% vs. 56.9, *p* = 0.022) and in T2b (68.2 vs. 48.5, *p* < 0.001). This suggest that the need for additional therapeutic modalities following surgical resection, and propose chemotherapy especially for the patients with LN metastasis as a more effective adjuvant strategy to treat any possible residual systemic tumor burden. In the present study, adjuvant chemotherapy was performed in roughly 30% of all patients and 47.1% of patients with LN metastasis. Although no standard postoperative adjuvant treatment or indications for treatment have been established, and in Korea adjuvant regimens were diverse, and based on 5-fluorouracil or gemcitabine, adjuvant chemotherapy was associated with improved overall survival for patients with LN metastasis in the present study. The GBC treatment with the least evidence is postoperative adjuvant therapy. To date, several clinical trials of adjuvant chemotherapy have been conducted worldwide [20,21,22,23]. However, most are hindered by small and heterogeneous patient populations that mix GBC with extra- and intrahepatic cholangiocarcinoma because of the low incidence of GBC. Hence, a multicenter, large-scale prospective study is warranted to validate the efficacy of adjuvant treatment strategies and to improve the prognosis of GBC.

There are several limitations in the present study. The most important limitation is that the tumor location was determined by radiologic evaluation, not by pathologic evaluation. The pathology of included patients could not be reviewed directly because the present study is retrospective, nationwide-multicenter study from the year of 2000 and it is not possible to obtain histologic slides or paraffin block from all patients. However, to overcome this limitation, the operative findings, pathologic reports and the images of computed tomography (CT) were collected separately from the registry data and were re-evaluated consistently by two surgeons (LSE, CYS) who were specialized to biliary pancreas surgery more than 15 years. When there was disagreement between two investigators, location was determined through consultation to radiologist. This was retrospective and nationwide multi-institutional study; therefore, the operative extent, and the policy for re-operation and indication of adjuvant chemotherapy varied among the institutions. Nevertheless, the information used was relatively precise because the participating surgeons and pathologists were specialized in hepatobiliary surgery and long-term follow-up data was available for most of the enrolled patients, which allowed for the analysis of survival and recurrence data as distinct characteristics from other nationwide studies. Notably, the completeness of the data was improved by the KOTUS-BP database registry data manager who visited the enrolled institutes and compiled the missing data.

## 5. Conclusions

The 5-year disease-specific survival rate for patients with T2a GBC was significantly better than that of T2b GBC. However, LN metastasis was only a significant poor prognostic factor in multivariate analysis. Given this finding and that systemic recurrence was more common, recurrence occurred more frequently in patients with LN metastasis, and adjuvant chemotherapy for the patients with LN metastasis improved survival, postoperative adjuvant chemotherapy should be considered particularly for the patients with LN metastasis. Furthermore, gallbladder bed resection is recommended for T2b GBC because ECx showed better survival trend than SCx with LN dissection in T2b GBC. Although the AJCC Eighth Edition subdivided the T2 stage of GBC intoT2a and T2b, the diagnosis, treatment, and prognosis of those stages remain debatable and further well-designed, large-scale studies are recommended.

## Figures and Tables

**Figure 1 jcm-09-03268-f001:**
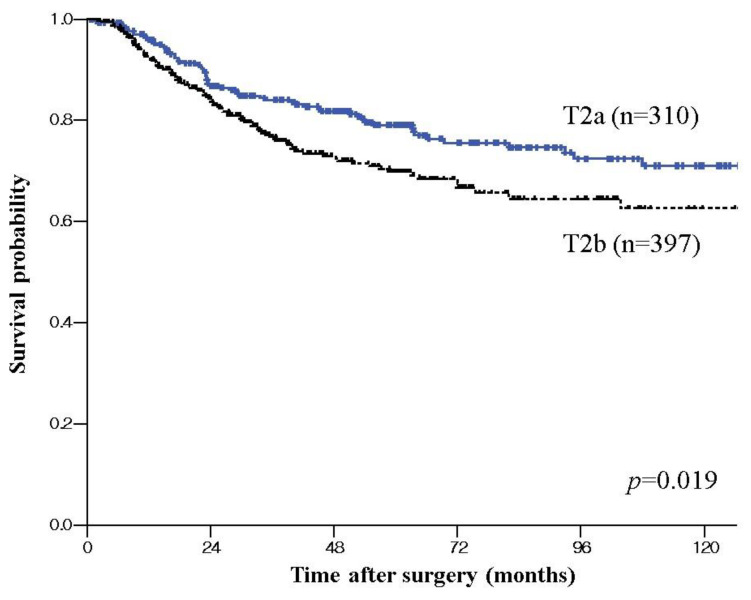
Disease specific survival curve of T2 gallbladder cancer (GBC) according to tumor location; Kaplan–Meier plots of patients with pT2a GBC showing significant better survival than that of the patients with T2b GBC.

**Figure 2 jcm-09-03268-f002:**
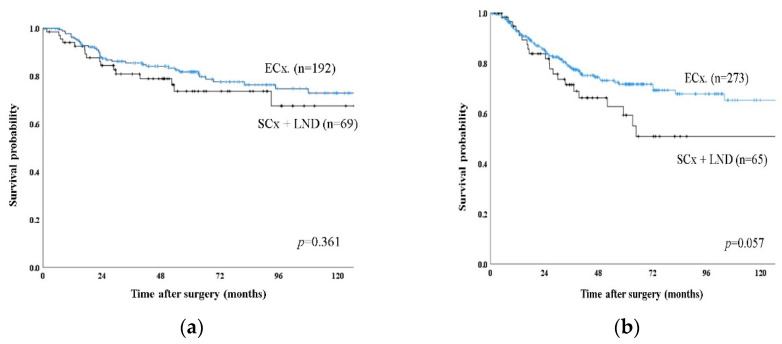
Disease specific survival curve of T2 GBC according to type of operation between extended cholecystectomy (ECx) and simple cholecystectomy with lymph node dissection (SCx + LND). (**a**) Kaplan–Meier plots of patients with T2a showing no difference in 5-year disease specific survival between ECx and SCx + LND. (**b**) Kaplan–Meier plots of patients with T2b showing a better survival trend without statistically significant in ECx than in SCx + LND.

**Table 1 jcm-09-03268-t001:** Clinicopathological characteristics of enrolled patients according to tumor location.

Characteristics	T2a (n = 310)	T2b (n = 397)	*p*-Value
Age (years)	66 ± 11	67 ± 11	0.892
Sex (M:F)	1:1.4	1:1.2	0.413
Combined Gallstone	45 (14.5%)	58 (14.6%)	0.922
Tumor size (cm)	2.98 ± 1.83	3.39 ± 2.15	0.027
Postoperative morbidity	31 (10.0%)	57 (14.3%)	0.107
Lymph node dissection	261 (84.2%)	338 (85.1%)	0.857
Lymph node metastasis	77 (29.5%)	128 (37.9%)	0.032
AJCC 8th N stage			0.031
N0	184 (70.5%)	211 (62.4%)	
N1	72 (27.6%)	109 (32.2%)	
N2	5 (1.9%)	18 (5.3%)	
Histologic differentiation			0.326
Papillary, well differentiation	116 (37.4%)	125 (31.5%)	
Moderate differentiation	131 (42.3%)	193 (48.6%)	
Poor differentiation	45 (14.5%)	56 (14.1%)	
Lymphovascular invasion (yes)	81 (36.5%)	105 (36.2%)	0.948
Perineural invasion (yes)	42 (19.7%)	103 (36.9%)	<0.001
Adjuvant chemotherapy (yes)	65 (24.9%)	119 (35.2%)	<0.001

**Table 2 jcm-09-03268-t002:** Recurrence patterns and sites according to tumor location.

Recurrence Site	Total (n = 707)	T2a (n = 310)	T2b (n = 397)	*p*-Value
Total	201 (28.4%)	70 (22.6%)	131 (33.0%)	0.006
Loco-regional	56 (29.0%)	15 (22.7%)	41 (31.3%)	
Liver bed	8 (14.3%)	1 (6.7%)	7 (17.1%)	
Common bile duct	17 (30.4%)	3 (20.0%)	14 (34.1%)	
Regional lymph node	23 (41.1%)	8 (53.3%)	15 (36.6%)	
Systemic	137 (71.3%)	51 (77.3%)	86 (68.7%)	
Liver	52 (38.0%)	23 (45.1%)	29 (33.7%)	
Lung	22 (16.1%)	5 (9.8%)	17 (19.8%)	
Peritoneal seeding	36 (26.3%)	15 (29.4%)	21 (24.4%)	
Para-aortic lymph node	18 (13.1%)	5 (9.8%)	13 (15.1%)	

**Table 3 jcm-09-03268-t003:** Multivariate analysis using the Cox regression proportional hazard model for disease specific survival in T2 GBC.

Variables	Univariate Analysis			Multivariate Analysis		
	Subgroup (n)	MST (mo)	*p*-Value	HR	95% CI	*p*-Value
T stage	pT2a (261)	52	0.02	1.340	0.919–1.955	0.128
	pT2b (338)	37				
Lymph node metastasis	Yes (206)	29	<0.001	3.222	1.960–4.489	<0.001
	No (393)	62				
Cellular differentiation	Well (193)	50	0.055	1.738	0–0.2967	0.995
	Moderate/poor (379)	40				
Lymphovascular invasion	Yes (186)	32	<0.001	1.721	0.505–5.858	0.117
	No (326)	50				
Perineural invasion	Yes (145)	31	<0.001	1.454	0.482–4.387	0.406
	No (347)	47				

MST, median survival time; HR, hazard ration; CI, confidence interval.

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
