# Peer review of "Prognostic Significance of Tumor Location in T2 Gallbladder Cancer: A Korea Tumor Registry System Biliary Pancreas (KOTUS-BP) Database Analysis"

_jcm, 2020, doi:10.3390/jcm9103268_

Round 1
Reviewer 1 Report
This manuscript is an original article that investigated clinical features and outcomes of T2 gallbladder cancer according to tumor location using the Korea tumor registry system-Biliary Pancreas database. The authors demonstrated that not tumor location but lymph node metastasis was a poor prognostic factor. They also stated that postoperative adjuvant chemotherapy should be considered for the patients with lymph node metastasis.
This study was conducted well, and the methods are appropriate. The data are presented clearly. In general, this is a well-written paper that presents interesting data. The results will be of interest to clinicians in the field.
The following minor issues require clarification:
- Several data shown in the Results section are not coincided with those described in the Discussion section (e.g. the rates of LN metastasis, recurrence rates, and survival rates).
- Please replace Figure 2a with Figure 2b.
- (P2L62-64) The description is overlapped with that in the Discussion section.
- No references can be seen.
Author Response
Journal of Clinical Medicine Oct 8, 2020
Dear pf. reviewer,
I appreciate the Journal of Clinical Medicine’s peer review of our manuscript, and your kind comment for revision. We have revised our manuscript according to the reviewers’ comments. The precise changes we have made are described on the next page and highlighted the changes in a revised manuscript.
We hope the changes satisfy the demand of the reviewers. It would be our great honor if this paper could be published in the Journal of Clinical Medicine.
Thank you in advance for your kind consideration. We look forward to hearing from you again.
Sincerely yours,
Jin-young Jang, M.D., Ph. D.
Department of Surgery and Cancer Research Institute
Seoul National University College of Medicine
101 Daehak-ro, Jongno-gu, Seoul
South Korea, 03080
Tel: 82-2-2072-2194 Fax: 82-2-766-3975
E-mail: jangjy4@snu.ac.kr
Point 1. Several data shown in the Results section are not coincided with those described in the Discussion section (e.g. the rates of LN metastasis, recurrence rates, and survival rates).
Response 1. Thank you for your kind comment and I’m so sorry for my carelessness. I matched the data in discussion with that in results and highlighted the data in yellow box. The details are as follows;
(Abstract, page 1, line reference 32) tumors (37.9% vs. 29.5%, p=0.032)
(Discussion, page 7, line reference 197) recurrence (33.0% vs. 22.6%, p=0.006)
(Discussion, page 7, line reference 200) tumor location (T2aN0 vs. T2bN0, 87.6% vs. 79.9%, p=0.190)
(Discussion, page 7, line reference 202) LN metastasis in pT2b GBC (37.9%)
(Discussion, page 7, line reference 202) pT2a GBC (29.5%, p=0.032).
Point 2. Please replace Figure 2a with Figure 2b.
Response 2. Thank you for your comment. I think there would be something wrong in the process of uploading the figures. I reversed figure 2a with figure 2b and to clarify, figure and figure legend were arranged in pairs as follows. (page 6). And according to second reviewer’s comment, I replaced marginal significance in figure 2-b legend with better survival trend.
Figure 2. Disease specific survival curve of T2 GBC according to type of operation between extended cholecystectomy (ECx) and simple cholecystectomy with lymph node dissection (SCx+ LND). a. Kaplan-Meier plots of patients with T2a showing no difference in 5-year disease specific survival between ECx and SCx+LND
Figure 2. Disease specific survival curve of T2 GBC according to type of operation between extended cholecystectomy (ECx) and simple cholecystectomy with lymph node dissection (SCx+ LND). b. Kaplan-Meier plots of patients with T2b showing showing a better survival trend without statistical significant in ECx than in SCx+ LND.
Point 3. (P2L62-64) The description is overlapped with that in the Discussion section.
Response 3. Thank you for your comment. As you pointed out, there are similar sentences about anatomical differences between peritoneal side and hepatic side of gallbladder in introduction and discussions section. However, they were not exactly same as follows; (P2L62-64) The hepatic side of the gallbladder has no serosa, is attached directly to the liver by loose connective tissue and has dense lymphatic, arterial and venous communications which allow it to easily invade the liver. Vs. (Discussion, P8L215-216) The T2b GBC is located on the hepatic side and can spread to the liver without penetrating the serosa.
And because the anatomical difference is thought to be an evidence to show different prognosis and to apply different surgical strategy according to tumor location and till now other evidence was not found, I think it could not be omitted or changed in the discussion. Please consider it.
Point 4. No references can be seen.
Response 4. Thank you for your kind comment. I think there would be something wrong in the process of uploading the manuscript. You can see references in new version of manuscript.
Reviewer 2 Report
Major points
- In the Abstract; Result; However, there was a marginally significant difference for T2b tumor (71.7% vs. 59.3%, 34 p=0.057).
This is a subjective opinion. In common, p<0.05 can be said as a significant difference. I do not know if "marginal significant difference" is a scientific word. Also, in conclusion, the word “marginally superior” sounds very subjective for me. What p-value do the authors define as having significance?
- In the Abstract; Result; Adjuvant chemotherapy was associated with improved survival for patients with lymph 35 node metastasis (69.1% vs. 49.1%, p=0.018).
The result contains the result of all T2 cancer. This is confusing. I think the research target is to compare T2a and T2b. Please make the whole story clear.
- Results; Fifty-five patients (18.5%) underwent a second operation 128 (liver resection with LN dissection) after SCx and there were no residual tumors at the gallbladder 129 bed in all 55 resected specimens.
This is interesting data. But, for T2b, simple cholecystectomy showed the tendency of worse prognosis compared to extended cholecystectomy although there is no significant difference (p<0.05). I think this should be discussed in the discussion part.
- Discussion; The most important limitation is that the tumor location was determined by radiologic evaluation, not by pathologic evaluation.
Why did the authors define the tumor location with pathological findings? This is not clear in the manuscript. The authors should describe the reason why they did not use pathological data if they think a pathological evaluation is better than radiological evaluation, according to their discussion part.
Minor points
- Results; Demographics: The mean patient age at diagnosis was 66±11 years (range 25-92).
I think mean age was 66 years old. “±11” means SD.
- Table 1;
What size does “size” mean? This means the size of gallbladder stone?
- Table 2;
The table is not well-constructed and looked busy. How about arranging with the catrgory of residual/ lymph/ distant metastases?
- Figure 2;
Please make clear which graph is 2a or 2b. I think two figures were opposite.
- Conclusion in the abstract looked confusing. Please make the story simple.
Author Response
Journal of Clinical Medicine Oct 8, 2020
Dear pf. reviewer,
I appreciate the Journal of Clinical Medicine’s peer review of our manuscript, and your kind comment for revision. We have revised our manuscript according to the reviewers’ comments. The precise changes we have made are described on the next page and highlighted the changes in a revised manuscript.
We hope the changes satisfy the demand of the reviewers. It would be our great honor if this paper could be published in the Journal of Clinical Medicine.
Thank you in advance for your kind consideration. We look forward to hearing from you again.
Sincerely yours,
Jin-young Jang, M.D., Ph. D.
Department of Surgery and Cancer Research Institute
Seoul National University College of Medicine
101 Daehak-ro, Jongno-gu, Seoul
South Korea, 03080
Tel: 82-2-2072-2194 Fax: 82-2-766-3975
E-mail: jangjy4@snu.ac.kr
<Major points>
Point 1. In the Abstract; Result; However, there was a marginally significant difference for T2b tumor (71.7% vs. 59.3%, 34 p=0.057).
This is a subjective opinion. In common, p<0.05 can be said as a significant difference. I do not know if "marginal significant difference" is a scientific word. Also, in conclusion, the word “marginally superior” sounds very subjective for me. What p-value do the authors define as having significance?
Response 1. Thank you for your very critical comment. We define p-value <0.05 as statistically significant value. In some papers, when p-value was slightly above 0.05, it was said marginally significant. However, strictly speaking, you’re right. So, I modified the manuscript as follows.
Previous
(in abstract, page 1, reference line 35) there was a marginal significant difference for T2b tumor (71.7% vs. 59.3%, p=0.057)
Revised
there was a better survival trend for T2b tumor after extended cholecystectomy (71.7% vs. 59.3%, p=0.057).
Previous
(in results, page 5, reference line 165) there was marginal significant difference for T2b tumor (71.7% vs. 59.3%,p=0.057)
Revised
there was a better survival trend without statistical significance for T2b tumor in ECx than in SCx+ LN dissection (71.7% vs. 59.3%, p=0.057).
Previous
(in conclusion, page 8, reference line 273) gallbladder bed resection is recommended for T2b GBC because ECx was marginally superior to SCx with LN dissection in T2b GBC
Revised
gallbladder bed resection is recommended for T2b GBC because ECx showed better survival trend than SCx with LN dissection
Point 2. In the Abstract; Result; Adjuvant chemotherapy was associated with improved survival for patients with lymph node metastasis (69.1% vs. 49.1%, p=0.018).
The result contains the result of all T2 cancer. This is confusing. I think the research target is to compare T2a and T2b. Please make the whole story clear.
Response 2. Thank you for your kind comment. As you mentioned, the above data means there is significant survival difference in all T2 GBC with LN metastasis according to receiving adjuvant chemotherapy or not. Although I didn’t show the data, I analyzed the effect of adjuvant chemotherapy on T2a group and T2b group respectively, and there was no survival gain in T2a, T2b, T2aN0, and T2bN0 groups. However, both T2a with LN metastasis (72.1% vs. 56.9, p=0.022) and T2b with LN metastasis (68.2 vs. 48.5, p<0.001) showed significant better survival after adjuvant chemotherapy comparing to no adjuvant chemotherapy. So, I showed the data combined T2a and T2b with LN metastasis (69.1% vs. 49.1%, p=0.018). But, as your comment, to show the difference between T2a and T2b group, I revised the manuscript as follows;
Previous
(Abstract, page 1, reference line 36)
Adjuvant chemotherapy was associated with improved survival for patients with lymph node metastasis (69.1% vs. 49.1%, p=0.018).
Revised
Adjuvant chemotherapy was associated with improved survival for patients with lymph node metastasis in T2a (72.1% vs. 56.9, p=0.022) and in T2b (68.2 vs. 48.5, p<0.001).
Previous
(Results, page 5, reference line 167)
Adjuvant chemotherapy was associated with improved survival for patients with lymph node metastasis (69.1% vs. 49.1%, p=0.018).
Revised
Adjuvant chemotherapy was associated with improved survival for patients with lymph node metastasis in T2a (72.1% vs. 56.9, p=0.022) and in T2b (68.2 vs. 48.5, p<0.001).
Point 3. Results; Fifty-five patients (18.5%) underwent a second operation (liver resection with LN dissection) after SCx and there were no residual tumors at the gallbladder bed in all 55 resected specimens.
This is interesting data. But, for T2b, simple cholecystectomy showed the tendency of worse prognosis compared to extended cholecystectomy although there is no significant difference (p<0.05). I think this should be discussed in the discussion part
Response 3. Thank you for your very critical comment. In general, residual tumors are identified about 20-60% after second operation in T2 GBC. However, our data showed that there were no residual disease after second operation. It is so difficult to explain this controversial result. There is selection bias in the present study because only 18.5% of patients underwent second operation and there was no consensual criteria for second operation in this retrospective multicenter study. So, I think well-designed prospective large-scale study with strict treatment algorithm is needed. Please consider this and understand that I added only one sentence as follows;
Revised
(Discussion, page 8, reference line 226)
Although in the present study, no residual disease was detected in 25 patients who re-operated for T2b GBC, there might be selection bias because only 18.5% of patients underwent second operation and there was no consensual criteria for second operation in this retrospective multicenter study.
Point 4. Discussion; The most important limitation is that the tumor location was determined by radiologic evaluation, not by pathologic evaluation.
Why did the authors define the tumor location with pathological findings? This is not clear in the manuscript. The authors should describe the reason why they did not use pathological data if they think a pathological evaluation is better than radiological evaluation, according to their discussion part.
Response 4. Thank you for your comment. I agree with you that this is the most important limitation of the present study. However, we could not review directly the pathology of included patients because the present study is retrospective, nation-wide multicenter study from the year of 2000 and we could not obtain slides or paraffin block. I added description as follows;
Revised
(Discussion, page 8, reference line 251)
The pathology of included patients could not be reviewed directly because the present study is retrospective, nationwide-multicenter study from the year of 2000 and it is not possible to obtain histologic slides or paraffin block from all patients.
Point 1. Results; Demographics: The mean patient age at diagnosis was 66±11 years (range 25-92).
I think mean age was 66 years old. “±11” means SD.
Response 1. Thank you for your comment. According to your comments, I revised as follows;
Revised
(in results, page 3, reference line 122). The mean patient age at diagnosis was 66 years (range 25-92)
Point 2. Table 1; What size does “size” mean? This means the size of gallbladder stone?
Response 2. Size in table 1 means tumor size. I modified table as follows;
(page 4, table1) Tumor size (cm)
.
Point 3. Table 2;. The table is not well-constructed and looked busy. How about arranging with the catrgory of residual/ lymph/ distant metastases?
Response 3. Thank you for your comment. I revised the table 2 as follows:
|
|
Total (n=707) |
T2a (n=310) |
T2b (n=397) |
P value |
|
|
||||
|
Total |
201 (28.4%) |
70 (22.6%) |
131 (33.0%) |
0.006 |
|
Loco-regional |
56 (29.0%) |
15 (22.7%) |
41 (31.3%) |
|
|
Liver bed |
8 (14.3%) |
1 (6.7%) |
7 (17.1%) |
|
|
Common bile duct |
17 (30.4%) |
3 (20.0%) |
14 (34.1%) |
|
|
Regional lymph node |
23 (41.1%) |
8 (53.3%) |
15 (36.6%) |
|
|
|
|
|
|
|
|
Systemic |
137 (71.3%) |
51 (77.3%) |
86 (68.7%) |
|
|
Liver |
52 (38.0%) |
23 (45.1%) |
29 (33.7%) |
|
|
Lung |
22 (16.1%) |
5 (9.8%) |
17 (19.8%) |
|
|
Peritoneal seeding |
36 (26.3%) |
15 (29.4%) |
21 (24.4%) |
|
|
Para-aortic lymph node |
18 (13.1%) |
5 (9.8%) |
13 (15.1%) |
|
Point 4. Figure 2; Please make clear which graph is 2a or 2b. I think two figures were opposite.
Response 4. Thank you for your comment. As you mentioned, figure 2a and figure 2b were reversed.
I think there would be something wrong in the process of uploading the figures. As your comment, I replaced figure 2a with figure 2b and to clarify, figure and figure legend were arranged in pairs as follows. (page 6).
Figure 2. Disease specific survival curve of T2 GBC according to type of operation between extended cholecystectomy (ECx) and simple cholecystectomy with lymph node dissection (SCx+ LND). a. Kaplan-Meier plots of patients with T2a showing no difference in 5-year disease specific survival between ECx and SCx+LND
Figure 2. Disease specific survival curve of T2 GBC according to type of operation between extended cholecystectomy (ECx) and simple cholecystectomy with lymph node dissection (SCx+ LND). b. Kaplan-Meier plots of patients with T2b showing showing a better survival trend without statistical significant in ECx than in SCx+ LND.
Point 5. Conclusion in the abstract looked confusing. Please make the story simple
Response 5. Thank you for your critical comment. According to your comment, I revised the manuscript as follows;
Previous
For T2 GBC, tumor location was not an independent prognostic factor. Extended cholecystectomy was marginally superior to simple cholecystectomy in T2b GBC. Lymph node metastasis was a significant poor prognostic factor. Hence, postoperative adjuvant therapy should be considered for the patients with lymph node metastasis.
Revised
(Abstract, page 1, reference line 40)
For T2 GBC, tumor location was not an independent prognostic factor. Lymph node metastasis was a significant poor prognostic factor and adjuvant chemotherapy should be considered for the patients with lymph node metastasis
Reviewer 3 Report
This is an important study assessing the differences in survival between T2a and T2b gallbladder cancer. Overall, the study is well written and the data are clearly reported. A large number of patients were reviewed for this study. Please assess the following comments for this manuscript:
#1- Materials and Methods: Tumor stages and locations were determined with the retrospective review of histopathological and imaging reports? Were preoperative imaging studies retrieved for all patients? Did lack of preoperative imaging for determination of tumor location lead to patients exclusion? Please clarify.
#2- No references are provided with the manuscript. For assessment of scientific validity or the reported data references should be added in the manuscript.
Author Response
Journal of Clinical Medicine Oct 8, 2020
Dear pf. reviewer,
I appreciate the Journal of Clinical Medicine’s peer review of our manuscript, and your kind comment for revision. We have revised our manuscript according to the reviewers’ comments. The precise changes we have made are described on the next page and highlighted the changes in a revised manuscript.
We hope the changes satisfy the demand of the reviewers. It would be our great honor if this paper could be published in the Journal of Clinical Medicine.
Thank you in advance for your kind consideration. We look forward to hearing from you again.
Sincerely yours,
Jin-young Jang, M.D., Ph. D.
Department of Surgery and Cancer Research Institute
Seoul National University College of Medicine
101 Daehak-ro, Jongno-gu, Seoul
South Korea, 03080
Tel: 82-2-2072-2194 Fax: 82-2-766-3975
E-mail: jangjy4@snu.ac.kr
Point 1. Materials and Methods: Tumor stages and locations were determined with the retrospective review of histopathological and imaging reports? Were preoperative imaging studies retrieved for all patients? Did lack of preoperative imaging for determination of tumor location lead to patients exclusion? Please clarify.
Response 1. Thank you for your comment. Tumor locations were determined with review of histopathological reports by specialized surgeon and review of CT images by specialized surgeon and specialized radiologist. Only patients whose CT images were available and tumor location could be determined by CT images were included in the present study. Lack of preoperative image for determination of tumor location were excluded. To clarify, we added the description as follows;
(page 3, line reference 94) The patients whose CT images were available and tumor location could be determined by these images were included in the present study
Point 2. No references are provided with the manuscript. For assessment of scientific validity or the reported data references should be added in the manuscript
Response 2. . Thank you for your kind comment. I uploaded manuscript with references. But. I think there would be something wrong in the process of uploading the manuscript. You can see references in new version of manuscript.
Round 2
Reviewer 2 Report
I believe the manuscript has been significantly improved and warranted publication in JCM. Thank you for the great revision and for providing detailed information.
Reviewer 3 Report
Thank you for considering the comments and suggestions for this manuscript. References are now correctly provided. The revision is otherwise satisfactory.